# Learning-Augmented Algorithms for Online Linear and Semidefinite Programming

**Elena Grigorescu**[*]
Purdue University, USA
elena-g@purdue.edu

**Young-San Lin**[*†]
University of Melbourne, Australia
nilnamuh@gmail.com

**Sandeep Silwal**[‡]
MIT, USA
silwal@mit.edu

**Maoyuan Song**[*§]
Purdue University, USA
song683@purdue.edu

**Samson Zhou**[¶]
UC Berkeley and Rice University, USA
samsonzhou@gmail.com

## Abstract

Semidefinite programming (SDP) is a unifying framework that generalizes both linear programming and quadratically-constrained quadratic programming, while also yielding efficient solvers, both in theory and in practice. However, there exist known impossibility results for approximating the optimal solution when constraints for covering SDPs arrive in an online fashion. In this paper, we study online covering linear and semidefinite programs, in which the algorithm is augmented with advice from a possibly erroneous predictor. We show that if the predictor is accurate, we can efficiently bypass these impossibility results and achieve a constant-factor approximation to the optimal solution, i.e., consistency. On the other hand, if the predictor is inaccurate, under some technical conditions, we achieve results that match both the classical optimal upper bounds and the tight lower bounds up to constant factors, i.e., robustness.

More broadly, we introduce a framework that extends both (1) the online set cover problem augmented with machine-learning predictors, studied by Bamas, Maggiori, and Svensson (NeurIPS 2020), and (2) the online covering SDP problem, initiated by Elad, Kale, and Naor (ICALP 2016). Specifically, we obtain general online learning-augmented algorithms for covering linear programs with fractional advice and constraints, and initiate the study of learning-augmented algorithms for covering SDP problems. Our techniques are based on the primal-dual framework of Buchbinder and Naor (Mathematics of Operations Research, 34, 2009) and can be further adjusted to handle constraints where the variables lie in a bounded region, i.e., box constraints.

## 1  Introduction

In the classical online model, an input is iteratively given to an algorithm that must make irrevocable decisions at each point in time, while satisfying a number of changing constraints and optimizing a

---

[*]E.G, Y.L, and M.S were supported in part by NSF CCF-1910659, NSF CCF-1910411, NSF CCF-2228814, and a Ross-Lynn Award.

[†]Work done while at Purdue University, USA.

[‡]Supported by an NSF Graduate Research Fellowship under Grant No. 1745302, NSF TRIPODS program (award DMS-2022448), and Simons Investigator Award.

[§]Supported in part by NSF CCF-2127806

[¶]Work supported by a Simons Investigator Award and by the National Science Foundation under Grant No. CCF-1815840, and done in part while at Carnegie Mellon University, USA.

fixed predetermined objective. A common metric for evaluating the quality of an online algorithm is the competitive ratio, which is the ratio between the "cost" of the algorithm and the best cost in hindsight, i.e., that of an optimal offline algorithm given the entire input sequence in advance. In the context of the minimization problems we study in this paper, an online algorithm is $c$-competitive if its cost is at most a multiplicative $c$ factor more than the cost of the optimal solution. Due to the irrevocable decisions, the changing constraints, or the number of possible different worst-case inputs, many online algorithms have undesirable competitive ratios that are impossible to improve upon without additional assumptions, e.g., [AAA$^+$09].

Due to advances in the predictive ability of machine learning models, a natural approach to overcome these computational barriers is to incorporate models with predictions, e.g., models that predict outcomes based on historical data. Unfortunately, due to the lack of provable worst-case guarantees, these predictions can be embarrassingly inaccurate when attempting to generalize to unfamiliar inputs, as shown in [SZS$^+$14], or simply not even satisfy the given constraints [BMS20]. Thus, rather than blindly following an erroneous machine learning predictor, recent focus has shifted to studying algorithms that use the output of these models as *advice*, and guarantee good competitive ratios both when the predictions are accurate, i.e., consistency, and when the predictions are poor, i.e., robustness.

Recently, [BMS20] studied the learning-augmented online set cover problem and related problems, using their linear programming (LP) formulation to incorporate additional advice through a primal-dual approach. One drawback of their seminal work, however, is that they assume both *integral* constraints, as well as integral advice, which restricts the modeling capabilities of the framework; it is natural to ask how an online algorithm can be improved when the advice is given in terms of probability distribution or some other meaningful *fractional* values. For example, for the online set cover problem, fractional advice can indicate how likely a set should be chosen instead of the binary decision of whether a set should be chosen or not; for the ski rental problem, the advice can be presented as a probability distribution over the total number of vacation days; in online network connectivity problems, the advice can indicate how likely an edge should be chosen.

In addition, linear programs cannot handle quadratic constraints and thus often fail to capture important aspects of fundamental optimization problems, which motivates the study of more general programs, such as semidefinite programming (SDP). SDP is a unifying framework that generalizes both linear programs and quadratically constrained quadratic programming (QCQP), while also yielding very efficient solvers, both in theory and in practice [VB96].

Before stating our contributions, we introduce some notation. For a learning-augmented problem, we are given a confidence parameter $\lambda \in [0, 1]$, where lower values of $\lambda$ denote higher confidence in the advice, and higher values denote lower confidence. An advice is a suggested solution for the online problem that is given. In the context of optimization problems including linear and semidefinite programming, a solution is a vector consisting of real numbers. We denote APX as the objective value of the online solution obtained by an online algorithm and compare it with (1) the objective value of the advice, denoted as ADV, and (2) the objective value of the offline optimal solution, denoted as OPT. The consistency and robustness of an online algorithm or solution for a minimization optimization problem are defined as follows.

**Definition 1.1.** An online solution with objective value APX is $C(\lambda)$-consistent if APX $\leq C(\lambda)$ADV. An online algorithm is $C(\lambda)$-consistent if it outputs a $C(\lambda)$-consistent solution for $C : [0, 1] \to \mathbb{R}_{\geq 1}$.

**Definition 1.2.** An online solution with objective value APX is $R(\lambda)$-robust if APX $\leq R(\lambda)$OPT. An online algorithm is $R(\lambda)$-robust if it generates an $R(\lambda)$-robust solution. Here, $R : [0, 1] \to \mathbb{R}_{\geq 1}$.

If an advice is accurate and we trust it, then we would like the solution to be close to the optimal, so ideally $C(\lambda) \to 1$ as $\lambda \to 0$. On the other hand, having $\lambda$ being close to 1 denotes no trust in the advice, so $R(1)$ should be close to the optimal competitive ratio of the best pure online algorithm.

## 1.1 Background and Contributions

We give a general paradigm for designing learning-augmented algorithms for online covering linear programming [BN09b], which generalizes the set cover problem [BMS20], as well as online covering semidefinite programming [EKN16], with possibly non-integral constraints and advice. Specifically, we present *primal-dual learning-augmented (PDLA)* algorithms for these problems, whose performance is close to the optimal offline solution when the advice is accurate, and also whose performance is asymptotically close to the optimal oblivious algorithm, if the advice is inaccurate.

Because of space constraints, we defer further discussions, additional related work, full algorithms, formal statements of theorems, and all the proofs to the full version of the paper.

Our PDLA algorithms consider the advice while approximately minimizing the objective value when the input arrives online. Our unifying paradigm applies to both online covering linear programs (LP) and online covering semidefinite programs (SDP) described below.

**Online covering linear programs.** A *covering* LP is defined as follows:

$$\text{minimize } c^T x \text{ over } x \in \mathbb{R}^n_{\geq 0} \text{ subject to } Ax \geq \mathbf{1}. \tag{1}$$

Here, $A \in \mathbb{R}^{m \times n}_{\geq 0}$ consists of $m$ *covering* constraints, $\mathbf{1}$ is a vector of all ones, and $c \in \mathbb{R}^n_{>0}$ denotes the positive coefficients of the linear cost function. In the online covering problem, the cost vector $c$ is given offline, and each of these covering constraints (rows) is presented one by one in an online fashion, that is, $m$ can be unknown. The goal is to update $x$ in a non-decreasing manner such that all the covering constraints are satisfied and the objective value $c^T x$ is approximately minimized.

The $O(\log n)$-competitive algorithm for online covering LPs presented in [BN09b] simultaneously solves both the primal covering LP (1) and the dual *packing* LP (2), defined as follows:

$$\text{maximize } \mathbf{1}^T y \text{ over } y \in \mathbb{R}^m_{\geq 0} \text{ subject to } A^T y \leq c. \tag{2}$$

The analysis in [BN09b] crucially uses LP-duality and strong connections between the two solutions to argue that they are both nearly optimal. The covering solution $x$ is an exponential function of the packing solution $y$ and both $x$ and $y$ are monotonically increasing. The problem naturally extends to the setting that relies on a *separation oracle* to retrieve an unsatisfied covering constraint where the number of constraints can be unbounded. However, as the framework in [BN09b] fixes all violating constraints, each arriving constraint might be slightly violated so that each individual fix may require a diminishingly small adjustment. Consequently, the algorithm may have to address exponentially many constraints. The framework was later modified in [GLQ21] which guarantees that addressing polynomially many constraints suffices.

In the learning-augmented problem, we are given a confidence parameter $\lambda \in [0,1]$ and $x' \in \mathbb{R}^n_{\geq 0}$ served as a fractional advice for LP (1). However, we do not have guarantees about the advice $x'$. More specifically, the objective value of the advice $c^T x'$ could be a horrendous approximation to the optimal objective value of LP (1) or $x'$ might not even satisfy the constraints.

We first show an efficient, consistent, and robust PDLA algorithm for the online covering LP (1). We use the condition number $\kappa$ to denote the upper bound for the ratio between the maximum positive entry and the minimum positive entry for each fixed column of $A$. For ease of presentation, we assume that $x'$ is *feasible*, i.e., there are no violating constraints caused by the advice $x'$.

**Theorem 1.3** (Informal). *Given a feasible advice $x' \in \mathbb{R}^n_{\geq 0}$ for LP (1) with confidence parameter $\lambda \in [0,1]$, there exists an $O\left(\frac{1}{1-\lambda}\right)$-consistent and $O\left(\log \frac{\kappa n}{\lambda}\right)$-robust online algorithm for the online covering LP problem that encounters polynomially many violating constraints.*

The formal version of Theorem 1.3 (see full version [GLS+22a]) also addresses the case when $x'$ is infeasible for LP (1). We note that Theorem 1.3 implies that when $\kappa = \text{poly}(n)$, the algorithm is $\log(n/\lambda)$-robust.

**Online covering semidefinite programs.** We generalize our approach for learning-augmented covering LPs to handle a more expressive family of optimization problems, namely, covering semidefinite programs. First, we introduce some standard notation. A matrix $A \in \mathbb{R}^{d \times d}$ is said to be *positive semidefinite* (PSD), i.e., $A \succeq 0$, if $v^T A v \geq 0$ for every vector $v \in \mathbb{R}^d$, or equivalently, all the eigenvalues of $A$ are non-negative. If $A$ is PSD and symmetric, then it is *symmetric positive semidefinite (SPSD)*. A partial order over SPSD matrices in $\mathbb{R}^{d \times d}$ can be induced such that $A \succeq B$ if and only if $A - B \succeq 0$. The setting of a covering SDP problem is as follows.

$$\text{minimize } c^T x \text{ over } x \in \mathbb{R}^n_{\geq 0} \text{ subject to } \sum_{j=1}^{n} A_j x_j \succeq B \tag{3}$$

where $A_1, \ldots, A_n \in \mathbb{R}^{d \times d}$ and $B \in \mathbb{R}^{d \times d}$ are SPSD matrices and $c \in \mathbb{R}^n_{>0}$.

In the online covering SDP problem introduced in [EKN16], we have the matrices $A_1, \ldots, A_n$ and the cost vector $c$ given offline. In each *round* $i \in [m]$ where $m$ can be unknown, we are given a new

SPSD matrix $B^{(i)}$ satisfying $B^{(i)} \succeq B^{(i-1)}$. The goal is to cover $B^{(i)}$ using a linear combination $x_1, \ldots, x_n$ of the matrices $A_1, \ldots, A_n$, so that $\sum_{j=1}^{n} x_j A_j \succeq B^{(i)}$, while minimizing the cost $c^T x$. Moreover, we must update $x$ in a non-decreasing manner, so that once some amount of the matrix $A_j$ is used in the covering at round $i$, then it must be used in all subsequent coverings in later rounds. The online covering SDP problem and its dual in round $i$ are as follows:

$$\text{minimize } c^T x \text{ over } x \in \mathbb{R}_{\geq 0}^n \text{ subject to } \sum_{j=1}^{n} A_j x_j \succeq B^{(i)} \tag{4}$$

$$\text{maximize } B^{(i)} \otimes Y \text{ over } Y \succeq 0 \text{ subject to } A_j \otimes Y \leq c_j \quad \forall j \in [n] \tag{5}$$

where $A \otimes B := \sum_{i,j} A_{i,j} B_{i,j} = \text{trace}(A^T B)$ is the Frobenius product.

We remark that the formulation of online covering SDP (4) generalizes online covering LP (1) when the constraint matrix is known offline but there is no guarantee which covering constraint (row) will arrive. In particular, the SDP formulation for online set cover with $n$ sets and $d$ elements all given offline (but without the knowledge of which elements arrive and their order) is the following: we define matrices $A_1, \ldots, A_n \in \{0, 1\}^{d \times d}$ where $A_j$ is a diagonal matrix whose diagonal is simply the indicator vector for the $j$-th set across the $d$ elements, i.e., entry $(k, k)$ of $A_j$ is 1 if and only if set $j$ contains element $k$. The matrices $B^{(i)}$ encode the variables that must be covered in round $i$, so that $B^{(0)}$ is the all zeros matrix and $B^{(i)} - B^{(i-1)}$ is the all-zeros matrix except with a single one in entry $(k, k)$ for the variable $k$ that must be newly covered in round $i$. Thus no online SDP algorithm can achieve competitive ratio $o(\log n)$ because even if fractional sets are allowed, no online algorithm can achieve competitive ratio better than $O(\log n)$ for the online set cover problem [BN09a].

An optimal $O(\log n)$-competitive online algorithm for covering SDPs was presented in [EKN16]. Similar to online covering LPs, an important idea in this line of work is to use weak duality and the strong connections between the primal and the dual solutions. Observe that if $x$ and $Y$ are feasible solutions for the primal and the dual, then $c^T x \geq \sum_{j=1}^{n} (A_j \otimes Y) x_j = \left( \sum_{j=1}^{n} (A_j x_j) \right) \otimes Y \geq B^{(i)} \otimes Y$, and hence the primal and the dual satisfy weak duality.

In the learning-augmented problem, we are given a confidence parameter $\lambda \in [0, 1]$ and a vector $x' \in \mathbb{R}_{\geq 0}^n$ that serves as advice. We have no guarantees about the advice, the objective value of the advice $c^T x'$ could be a terrible approximation or $x'$ might not even be feasible.

We use $\kappa$ to denote the ratio of the largest positive eigenvalue to the smallest positive eigenvalue of the matrices $A_1, \ldots, A_n, B^{(1)}, \ldots, B^{(m)}$ and achieve the following.

**Theorem 1.4** (Informal). *Given a feasible advice $x' \in \mathbb{R}_{\geq 0}^n$ for SDP (4) with confidence parameter $\lambda \in [0, 1]$, there exists a polynomial time, $O\left(\frac{1}{1-\lambda}\right)$-consistent, and $O\left(\log \frac{\kappa n}{\lambda}\right)$-robust online algorithm for the online covering SDP problem.*

The formal version of Theorem 1.4 also addresses the case when $x'$ is infeasible for SDP (4). It implies that we can achieve a constant factor approximation to the optimal solution when the advice is accurate ($O(1)$-competitive), which breaks the known $\Omega(\log n)$ competitive ratio obtained by the oblivious online algorithm for covering SDP in [EKN16]. Moreover, for $\kappa = \text{poly}(n)$, we match the optimal approximation ratio of $O(\log n)$ up to constants when the advice is arbitrarily bad.

**Adding box constraints.** In both the LP and SDP case, it is natural to have the requirement that the variables must lie in a bounded region. We extend our results for both online covering LPs and online covering SDPs to this case where each coordinate $x_j$ of the variable vector $x$ is required to lie in the interval $[0, u_j]$. We achieve qualitatively similar consistency and robustness trade-offs as before which are summarized in Table 1. Instead of $\kappa$, a sparsity term $s$ now appears in the robustness portion of the competitive ratio. In the LP case for set cover, $s$ exactly captures the row sparsity (i.e., the maximum number of non-zero entries of any row).

**Theoretical applications.** We emphasize that our framework uses a continuous approach that is amenable to other learning-augmented optimization problems and supports fractional advice, which may be interpreted as probabilities. For example, as in [EKN16, AW02, WX06], our framework for covering SDPs may be applied to the quantum hypergraph covering problem. In the full version paper [GLS+22a], we apply our PDLA algorithm for covering LPs with box constraints in order to

| Paper | Problem | Approximation Guarantee | Approach |
|---|---|---|---|
| [BN09a] | online covering LP | with and without box constraints: $O(\log n)$-competitive | continuous guess-and-double |
| [EKN16] | online covering SDP | without box constraints: $O(\log n)$-competitive with box constraints: $O(\log s)$-competitive | continuous guess-and-double efficient updating |
| [BMS20] | learning-augmented online set cover | without box constraints: $O(1/(1-\lambda))$-consistent $O(\log(d/\lambda))$-robust | discretized |
| [GLQ21] | online covering LP | without box constraints: $O(\log n)$-competitive | continuous guess-and-double efficient updating |
| **This Work** | learning-augmented online covering LP and SDP with fractional advice | without box constraints: $O(1/(1-\lambda))$-consistent $O(\log(\kappa n/\lambda))$-robust with box constraints: $O(1/(1-\lambda))$-consistent $O(\log(s/\lambda))$-robust | continuous guess-and-double efficient updating |

Table 1: Summary of the competitive, consistency, and robustness ratios. We assume that the advice is feasible for the learning-augmented problems. Here, $n$ refers to the number of sets or variables, $\lambda \in [0, 1]$ refers to the confidence parameter, $\kappa$ refers to the condition number, $d$ refers to row sparsity, and $s$ refers to sparsity. We note that online covering LP with box constraints generalizes online set cover with $s = d$. In [EKN16], the guess-and-double scheme is not used for online SDP covering with box constraints.

obtain online algorithms for: (1) the fractional online set cover problem with fractional advice, and for (2) the online group Steiner tree problem on trees, where a min-cut algorithm is used as a separation oracle to retrieve violating constraints. Our learning-augmented solver for the group Steiner problem on trees can be employed as a black-box for other related problems, including group Steiner tree on general graphs, multicast problem on trees, and the non-metric facility location problem [AAA+06].

## 1.2 Subsequent developments

Subsequent to our work, a significantly simpler algorithm with tighter qualitative guarantees was brought to our attention by Roie Levin. We describe the algorithm in the full version of the paper [GLS+22b], with his permission. Nevertheless, we expect that the techniques and analysis that we introduce in this paper may be of independent interest for other related problems or settings, such as the advice being adaptive, or in settings of multiple experts. We believe that understanding the full power of the techniques developed in this paper is an intriguing direction for further research in the still emerging area of learning-augmented algorithms.

## 1.3 Overview of our Techniques

We now give a technical overview of our algorithms and describe how both our algorithms for covering LPs and SDPs are guided by several common underlying principles.

**Previous approaches.** A natural starting point would be the PDLA algorithm for online set cover by [BMS20], who adapted the primal-dual approach in [BN09a] to incorporate external advice. We recall that in the covering LP formulation of the online set cover problem, each row denotes an element and each column denotes a set. The constraint matrix has entries that are either $0$ or $1$. An entry is $1$ if and only if the element (row) belongs to the set (column). Additionally, for the online set cover problem considered in [BMS20], each set is either included in the advice or not, i.e., each coordinate of the suggested indicator vector for the set selection is either $1$ or $0$. While it seems plausible that one could extend the *discretized* approach of [BMS20] to handle general coefficients in the constraint matrix, i.e., the online covering LP problem, it is unclear how the growth rates of

For each update, while there exists a violating constraint:

1. Determine a violating constraint.

2. Acquire a "growth rate" for each variable depending on its coefficient in the violating constraint, the corresponding cost, and the advice.

3. Use a guess-and-double approach to determine how fast each of the variables are increased by their growth rates.

4. Increase the variables continuously until the constraint is satisfied.

Figure 1: Summary of our framework

the variables can be adjusted to guarantee dual feasibility. This is because the positive coefficients in every covering constraint (all with value 1) are *balanced* in the online set cover problem, which turns out to be a crucial ingredient to argue dual feasibility by the discretized approach, but we do not have this guarantee for general covering LPs with arbitrary positive coefficients. Instead, we use a different framework inspired by the classical online algorithm literature, e.g., [BN09b, EKN16]. We present a summary of our framework in Figure 1 and describe it in more details below.

**Continuous updates.** Each time a new constraint arrives, we *continuously* increase the variables until the constraint is satisfied. We adjust this growth rate of each variable based on its cost in the objective linear function, its coefficient in the arriving constraint, and the advice: a variable is increased at a slower rate if its cost is more expensive, its coefficient in the constraint has a smaller value, or it is less recommended by the predictor. The introduction of fractional values in the advice is the main technical obstacle in our setting. In particular, our algorithm must behave differently in the case where a variable has not reached the fractional value recommended by the advice compared to the case where it has reached the recommended value, but the solution does not satisfy all constraints. By contrast, in the integral advice setting of [BMS20], the recommendation value always coincides with the limit at 1. To this end, once the variable reaches the recommended value, our algorithms judiciously decelerate the growth of the variable.

**Guess-and-double.** However, by allowing the coefficients of the constraint matrix to be arbitrary, the optimal objective value OPT can be arbitrary and we need a nice estimate for this. Thus, we adopt the *guess-and-double* technique, e.g., [BN09b, EKN16, GLQ21], where the algorithm is executed in *phases*, so that in each phase we propose a lower-bound estimate of OPT, and the algorithm enters the next phase when the value exceeds our estimate. Note that such techniques are not necessary for [BMS20], as their assumption of coefficients in $\{0, 1\}$ implicitly provided bounds on OPT.

**Efficient updating.** In more general applications, each arriving update may induce a large or even infinite number of constraints, such as an infinite number of directions induced by an SDP constraint. But now if we sequentially choose a violating constraint and satisfy the constraint exactly as in [BMS20], then there is no guarantee that we will satisfy all the constraints in a small number of iterations. Thus, another technique we adopt to ensure efficiency in conjunction with the guess-and-double technique is to satisfy each arriving constraint by a factor of 2. That is, we instead continue to increment the primal variables until the violating constraint is satisfied by a factor of 2, which ensures that at least one primal variable is doubled, which also implies a polynomial upper bound on the number of violating constraints that must be considered.

**Showing robustness and consistency.** With the introduction of general coefficients within many components of our LP formulation, the robustness analysis in [BMS20] is no longer applicable, so instead we adapt the primal-dual analysis in [BN09b] for general covering LP problems. In particular, we deal with the general coefficients via a delicate telescoping argument for dual feasibility, since we tune and change the growth rates multiple times even within the same phase. Towards obtaining the consistency bound, we partition the growth rate based on whether the variable has exceeded the value in the advice, and argue that the growth rate not credited to the advice is at most a certain factor of the growth rate credited to the advice, similar to the line of the argument presented in [BMS20].

**Extending to online covering SDPs with advice.** We now have arriving matrices rather than arriving elements, so that at each time we need to cover a new PSD matrix $B^{(i)}$ that can be *larger* than the previous PSD matrix $B^{(i-1)}$ in an infinite number of directions. We repeatedly look at the direction with the largest mass that needs to be covered, i.e., the largest eigenvector $v$ of $B^{(i)} - \sum_{j=1}^{n} A_j x_j$.

Then to cover the direction $v$, we set the growth rate of the coefficient of each matrix $A_j$ proportional to the amount that the matrix *aligns* with $v$, i.e., proportional to $v^T A_j v = A_j \otimes V$, where $V = vv^T$ and $\otimes$ is the Frobenius product. Unfortunately, it does not suffice to cover $v$ alone – there may be many other directions for which $B^{(i)} - \sum_{j=1}^n A_j x_j$ is not covered. However, as we satisfy the violating constraint by a factor of 2, the amount of vectors we have to cover is similarly upper-bounded as in the aforementioned approach for the online covering LP problem. Lastly, we remark that our unifying framework can be naturally applied to any online problem that has a covering LP or SDP formulation, equipped with a fractional advice and a confidence parameter.

## 2 PDLA Algorithms for Online Covering LPs and SDPs

We describe our PDLA algorithms for online covering LPs and SDPs and sketch the proof for their guarantees. For covering LPs (1), we have $A \in \mathbb{R}_{\geq 0}^{m \times n}$ with $m$ covering constraints, $\mathbf{1}$ is a vector of all ones, and $c \in \mathbb{R}_{>0}^n$ denotes the positive coefficients of the cost function. We use $A_i$ to denote row $i$ of $A$ and $a_{ij}$ to denote the entries of $A$, and $c_j$ to denote the $j$-th entry of $c$. For the covering SDP (4), we have SPSD matrices $A_j$'s and $B^{(i)}$'s. For both covering LPs and SDPs, the covering constraints arrive online and $c$ is given offline. The goal is to update $x$ in a non-decreasing manner such that $x$ is feasible and the objective $c^T x$ is minimized. We are given an advice $x' \in \mathbb{R}_{\geq 0}^n$.

We use a guess-and-double approach. The algorithms work in *phases*. Let $\mathsf{OPT}$ be the optimal objective value of LP (1) or SDP (4). We estimate a lower bound $\alpha(r)$ for $\mathsf{OPT}$ in phase $r$. In phase 1, let $\alpha(1) \leftarrow \min_{j \in [n]} \{c_j / a_{1j}\}$ be a proper lower bound for $\mathsf{OPT}$ in LP (1) ($\alpha(1) \leftarrow \min_{j \in [n]} \{c_j \operatorname{trace}(B^{(1)}) / \operatorname{trace}(A_j)\}$ in SDP (4)). For each subsequent phase, $\alpha(r+1) \leftarrow 2\alpha(r)$.

In the beginning of phase $r$, $x_j^{(r)} \leftarrow \min\{x_j', \alpha(r)/(2nc_j)\}$. If $x_j' \leq \alpha(r)/(2nc_j)$, then it is possible that $\alpha(r)/(2nc_j)$ is large, so we have to set $x_j^{(r)} = x_j'$ to ensure consistency. On the other hand, if $x_j' \geq \alpha(r)/(2nc_j)$, then it is possible that $x_j'$ is large and the advice is bad, so we have to set $x_j^{(r)} = \alpha(r)/(2nc_j)$ to ensure robustness.

Whenever we have a new constraint $i$ in LP (1), we introduce a new dual variable $y_i \leftarrow 0$. For SDP (4), the dual variable $Y$ is reset to zero matrix in a new phase. Once the online objective in phase $r$ exceeds $\alpha(r)$, we proceed to the next phase $r+1$ from the current constraint (let us call it constraint $i_{r+1}$, in particular, $i_1 = 1$). The purpose of $\alpha(r)$ is to accelerate the growth rate of each variable $x_j^{(r)}$ so that a violating constraint can be quickly covered without incurring a high cost for the objective. For a violating constraint $i$, the growth rate of each variable $x_j^{(r)}$ is then scaled proportional to $a_{ij}$ for covering LPs ($A_j \otimes V$ for covering SDPs) and inversely proportional to $c_j$, so that the variables that are more aligned with the violating constraint in the LP (implicit violating linear constraint induced by $V$ in the SDP) are valued more and the variables that are more expensive are valued less. If the advice does not adequately cover constraint $i$, then we increase each variable according to its growth rate until the (implicit) constraint is satisfied by a factor of 2. Otherwise if the advice covers constraint $i$, then before we increase each variable, we further adjust the growth rate of each variable based on a combination of the suggested weight by the advice and our confidence in the advice. Here we use an indicator function where $\mathbf{1}_{x_j^{(r)} < x_j'} = 1$ if $x_j^{(r)} < x_j'$ otherwise $\mathbf{1}_{x_j^{(r)} < x_j'} = 0$. It should be noted that since $x$ must be updated in a non-decreasing manner, the algorithm maintains $\{x_j^{(\ell)}\}_{\ell \in [r]}$, which denotes the value of each variable $x_j$ from phase 1 to phase $r$, and the value of each variable $x_j$ is set to $\max_{\ell \in [r]} \{x_j^{(\ell)}\}$. We defer the phase scheme to the full version paper [GLS+22a]. We describe the continuous primal-dual approach in phase $r$ and round $i$ in Algorithms 1 and 2. We note that the dual variables are used as the proxy to increase the primal variables. The increment functions are exponential so that the primal variables are increased in terms of their growth rates. The coefficients $B_j$ and $D_j$ are used to match the boundary conditions.

We remark that although we would like to increase the variables in a continuous fashion, we can nevertheless implement our algorithm in a discrete manner for any desired precision by using binary search. For covering LPs, the approach of satisfying each arriving violating constraint by a factor of 2 guarantees that the number of iterations is polynomially upper-bounded. This implies efficient

applications on problems that generate covering LPs with exponentially many or unbounded number of constraints, where violating constraints are retrieved by a separation oracle.

**Proof Sketch for Theorem 1.3.** We analyze Algorithm 1. Let $P(r) = c^T x$ and $D(r) = \mathbf{1}^T y$ be the primal covering and the dual packing objective in phase $r$, respectively. To show robustness, we show that the following claims are satisfied: (1) $x$ is feasible for LP (1), (2) for each *finished* phase $r$, $\alpha(r) \le 6D(r)$, (3) $y/\Theta\left(\log \frac{\kappa n}{\lambda}\right)$ is feasible for LP (2), and (4) the covering objective satisfies $c^T x \le 2\alpha(r')$, where $r'$ is the last phase.

Claim (1) follows by the termination condition at line 9 and the phase scheme. Claim (2) follows by taking the partial derivatives of the increment function and analyzing the accumulative primal objective value. Claim (3) follows by a careful telescoping argument. Claim (4) follows by observing that the sum of the objective value of all phases does not exceed $2\alpha(r')$ where $r'$ is the last phase.

We can then deduce that $c^T x \le \Theta(1)\alpha(r') \le \Theta(1)D(r') \le O\left(\log \frac{\kappa n}{\lambda}\right) \mathsf{OPT}$ where the first inequality follows by claim (4), the second inequality follows by claim (2), and the last inequality follows by claim (3) and weak duality, i.e., the objective value of any feasible packing solution for LP (2) is upper-bounded by $\mathsf{OPT}$.

To show consistency, upon the arrival of constraint $i$ in phase $r$, we increment $x^{(r)}$ in terms of $y_i$ and decompose $P(r)$ into two parts: $P_c$, the contribution from the advice, and $P_u$, the contribution from the online algorithm. The *increase* of $P(r)$ is also decomposed, i.e., $\frac{\partial P(r)}{\partial y_i} = \frac{\partial P_c}{\partial y_i} + \frac{\partial P_u}{\partial y_i}$. We initialize $P_c$ to a non-negative value and $P_u = 0$ in the beginning of phase $r$, show that $\frac{\partial P_u}{\partial y_i} \le \frac{2+\lambda}{1-\lambda} \frac{\partial P_c}{\partial y_i}$, and ultimately conclude that $P(r) = P_c + P_u \le O(\frac{1}{1-\lambda})P_c$. Since the algorithm increments $x^{(r)}$ until the violating constraint is satisfied by a factor of 2, the growth of $x^{(r)}$ is sufficient so that we only have polynomially many constraints to address.

**Proof Sketch for Theorem 1.4.** We analyze Algorithm 2. The proof is analogous to the proof of Theorem 1.3. The difference is that now we have $D(r) = B^{(i)} \otimes Y$. In the third claim of the proof of robustness, we instead show that $Y/\Theta\left(\log \frac{\kappa n}{\lambda}\right)$ is feasible for SDP (5). For consistency, we increment $x^{(r)}$ in terms of $Y$, where the increase of $Y$ is $\delta V$ for $\delta = 0$ in the beginning. Here we increment $\delta$ and consider the derivatives of $P(r)$, namely $P_c$, and $P_u$ w.r.t. $\delta$ instead of $y_i$.

| **Algorithm 1** PDLA Online Covering LP | **Algorithm 2** PDLA Online Covering SDP |
|---|---|
| 1: **for** each $j \in [n]$ **do** | 1: **while** $\sum_{j=1}^{n} A_j x_j^{(r)} \not\succeq B^{(i)}$ **do** |
| 2:    **if** $A_i x' \ge 1$ **then** | 2:    Find SPSD matrix $V$: $\sum_{j=1}^{n}(A_j x_j^{(r)}) \otimes V < B^{(i)} \otimes V$. |
| 3:      $D_j \leftarrow \frac{\lambda}{A_i \mathbf{1}} + \frac{(1-\lambda)x_j' \mathbf{1}_{x_j^{(r)} < x_j'}}{\sum_{k=1}^{n} a_{ik}x_j' \mathbf{1}_{x_j^{(r)} < x_j'}}$ | 3:    **for** each $j \in [n]$ **do** |
| | 4:      **if** $\sum_{k=1}^{n} A_k x_k' \succeq B^{(i)}$ **then** |
| 4:    **else** | 5:        $D_j \leftarrow \frac{\lambda B^{(i)} \otimes V}{\sum_{k=1}^{n} A_k \otimes V} + \frac{(1-\lambda)x_j' \mathbf{1}_{x_j^{(r)} < x_j'} B^{(i)} \otimes V}{\sum_{k=1}^{n} \mathbf{1}_{x_k^{(r)} < x_k'} A_k x_k' \otimes V}$, |
| 5:      $D_j \leftarrow \frac{1}{A_i \mathbf{1}}$. | |
| 6:    $Y_j^{(i-1)} \leftarrow \sum_{k=i_r}^{i-1} a_{kj} y_k$. | 6:    **else** |
| 7:    $B_j \leftarrow \frac{x_j^{(r)} + D_j}{\exp\left(\left(Y_j^{(i-1)} + a_{ij}y_i\right)/c_j\right)}$. | 7:        $D_j \leftarrow \frac{B^{(i)} \otimes V}{\sum_{k=1}^{n} A_k \otimes V}$. |
| 8: **if** $A_i x^{(r)} < 1$ **then** | 8:    $B_j \leftarrow \frac{x_j^{(r)} + D_j}{\exp\left(\frac{A_j \otimes Y}{c_j}\right)}$. |
| 9:    **while** $A_i x^{(r)} < 2$ **do** | |
| 10:      **for** each $j \in [n]$ **do** | 9: **while** $\sum_{j=1}^{n} A_j x_j^{(r)} \otimes V < 2B^{(i)} \otimes V$ **do** |
| 11:        Increase $y_i$ continuously. | 10:    **for** each $j \in [n]$ **do** |
| 12:        Increase $x_j^{(r)}$ simultaneously by | 11:      Set $\delta = 0$ and increase it continuously. |
| | 12:      Increase $Y$ by continuously adding $V\delta$ to $Y$. |
| $$x_j^{(r)} \leftarrow B_j \exp\left(\frac{Y_j^{(i-1)} + a_{ij}y_i}{c_j}\right) - D_j.$$ | 13:      Increase $x_j^{(r)}$ simultaneously by |
| | $$x_j^{(r)} \leftarrow B_j \exp\left(\frac{A_j \otimes Y}{c_j}\right) - D_j.$$ |
| 13:      **if** any $x_j^{(r)}$ reaches $x_j'$ **then** | |
| 14:        Break and go to line 1. | 14:    **if** any $x_j^{(r)}$ reaches $x_j'$ **then** |
| | 15:      Break and go to line 3. |

# 3 Empirical Evaluations

We demonstrate the applicability of our algorithmic framework on a synthetic and real datasets. Our focus will be on online covering algorithms with fractional hints and entries. We focus on this setting since it is the simplest of our algorithms and already captures key points of the overall framework. Note that prior work [BMS20] has already demonstrated the empirical benefit of learning-based methods for online covering with integral hints/constraints, albeit on synthetic datasets.

**Datasets.** For our synthetic dataset, the constraint matrix $A$ represents a $n \times n$ matrix where each entry is uniformly in $\{0, 1\}$ and whose rows arrive online. We set $n = 500$. The objective function $c$ is a scaled vector with entries uniform in $[0, 1]$. Our graph dataset is constructed as follows. We have a sequence of nine (unweighted) graphs which represents an internet router network sampled across time [LK14, LKF05][6]. The graphs have approximately $n \sim 10^4$ nodes and $m \sim 2.2 \cdot 10^4$ edges. We note that the nodes of the graphs are labeled and the labeling is consistent throughout the different time stamps. Each graph defines an instance of the set cover problem derived from the standard vertex cover to set cover reduction [Kar72]. The objective function $c$ will be the same as the synthetic case so our problem represents an instance of weighted vertex cover. All experiments are done in a 2021 M1 Macbook Pro with 32 gigabytes of RAM and implemented in Python 3.9.

**Predictions.** First we describe predictions for the synthetic dataset. We consider 2 types of predictions: (1) first find the optimal offline solution $x$ by solving the full linear program and then noisily corrupt the entries of $x$ by setting the entries to be 0 independently with probab. $p$. This is the same prediction used in [BMS20] which is referred as 'replacement rate' strategy. (2): the motivation of these predictions is the following: we are solving many related problem instances. To mimic this, we have matrices $A_0, A_1, \ldots$ where each index represents a new problem instance. $A_0$ is our synthetic matrix and $A_{i+1}$ updates $A_i$ by flipping $n$ entries at random. We fix $c$ to be the same. The predictions for all instances $i \geq 1$ are given by the optimal offline solution generated from the first instance $A_0$. This "batch" experimental design naturally models the scenario where the current problem instance is similar to past instances and so one can hope to utilize past learned information. A similar style of predictions, although not in an online context, has been employed in [CEI+22, EFS+22, DIL+21, CSVZ22]. For our graph dataset, we first solve the set cover instance on the 1st graph in the family using an offline algorithm. We then use the solution from the 1st graph as the hint for all subsequent graphs. We also noisy alter this hint for one of our experiments using the replacement rate strategy. Note that the set of vertices might vary across graphs. In this case, we set the corresponding entry in the hint vector to be 0.

**Results.** Figure 2 shows the results on the synthetic dataset while Figure 3 refers to our graph dataset. First, Figure 2a considers a single online instance of the synthetic dataset. Our prediction is the offline optimal solution. It shows a smooth trade-off in the competitive ratio as the parameter $\lambda$ ranges from 0 (full trust in the predictions) to 1 (no hints), as predicted by our theoretical bounds. Since the instance is random, we plot the average of 20 trials for each setting of $\lambda$ and show 1 standard deviation. The plot validates the consistency of our algorithms as the competitive ratio is a *factor of 2* lower with accurate predictions. In contrast, Figure 2b validates the robustness of our algorithm. There we consider the "replacement rate" strategy and randomly zero out the entries of the prediction (which is again the offline optimum) independently. The expected fraction of entries in the hint vector being set to zero is denoted as the corruption rate and is shown in the $x$-axis. We see that for a fixed setting of $\lambda$, such as $\lambda = 0.1$, our algorithm performs much better with hints than without when the corruption factor is low. However, as we increase the corruption, the performance of the algorithm degrades. Crucially, the performance *does not* degrade arbitrarily worse compared to $\lambda = 1$ (no hints) performance and our algorithm with hints is able to outperform the baseline up to a high corruption factor. In Figure 2c, we consider a "batch" experimental design for our synthetic dataset with 20 time steps. The green curve shows the competitive ratio without using any hints, i.e., $\lambda = 1$ (the baseline). The orange curve shows the competitive ratio across the varying instances when we use a batch prediction. The blue curve showcases more powerful predictions where the offline optimal of time step $t - 1$ is used as the hint for time step $t$. We display the average values across 20 instances. As the time step increases, the orange curve drifts upwards, which is intuitive as the problem instances are increasingly different. Nevertheless, the batch hint stays valid for many time steps. As expected, the blue curve consistently has the lowest competitive ratio as the hints are also updated.

---

[6]Graphs can be accessed in `https://snap.stanford.edu/data/Oregon-1.html`

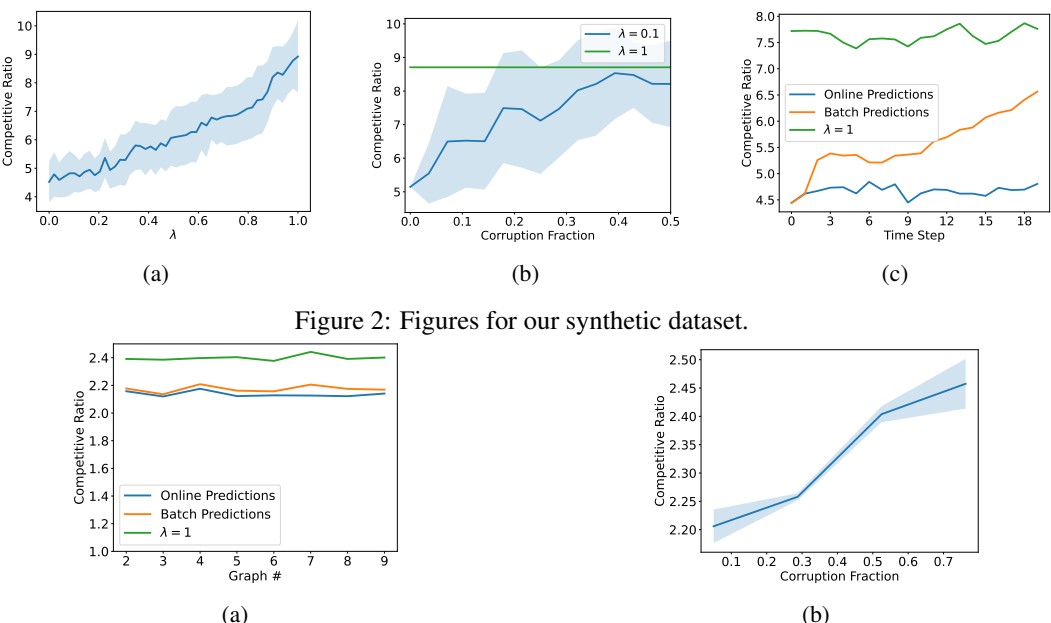

Figure 2: Figures for our synthetic dataset.

Figure 3: Figures for our time-varying graph dataset.

We now describe results on our graph dataset, given in Figure 3. In Figure 3a, the green curve represents not using any hints (baseline) while the orange curve shows the competitive ratio as we vary the graph instance while using the hint derived from graph #1 ($\lambda$ is set to $0.1$). It is shown that the hints help outperform the baseline and in addition, the hints stay accurate even if the structure of graph #9 has drifted away from that of graph #1. The online predictions, shown in blue, does marginally better than the batch version. Figure 3b shows a similar plot as Figure 2b. We consider a replacement rate strategy where we zero out coordinates of the hint vector independently with varying probabilities. The curve shows the average of $5$ trials and $\lambda = 0.1$ again. The same qualitative message as Figure 2b holds: while the corruption rate is small, we achieve a similar competitive ratio as in Figure 3a and as the corruption rate increases, there is a smooth increase in the competitive ratio.

In addition to extending and complementing the experimental results of [BMS20], we summarize our experimental results in the following points: (a) Our theory is predictive of experimental performance and qualitatively validates our robustness and consistency trade-offs. (b) Our algorithm framework which underlies all of our algorithm contributions is efficient to carry out and execute in practice. (c) Learning-augmented online algorithms can be applied to real world datasets varying over time such as in the analysis of graphs derived from a dynamic network.

**Open Problems.** As stated in [BMS20], a natural future direction is to design PDLA algorithms for packing LPs. For general online packing LPs, an $O(1/\log \kappa)$-competitive online solution can be obtained only if a condition number $\kappa$ is known offline [BN09b], implying impossibility results without assumptions. The hope here is to study structured packing problems, e.g., load balancing [BN09a] and ad-auction revenue maximization [BJN07].

# Acknowledgment

We thank Roie Levin for bringing to our attention the significantly simpler algorithm described in the full version of the paper [GLS$^+$22b].

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
