# OpenReview forum: "Learning-Augmented Algorithms for Online Linear and Semidefinite Programming"
_NeurIPS.cc/2022/Conference — NeurIPS 2022 Accept_

### Official Review · Reviewer_LaFt · 2022-07-10

**Rating:** 6
**Confidence:** 4
**Soundness:** 3 good
**Presentation:** 2 fair
**Contribution:** 4 excellent

**Summary:**

The paper considers online linear and semidefinite programming problem and makes use of the predicted optimal solutions to give algorithms. The authors extend the primal-dual learning-augmented framework of online set cover (NeurIPS 2020) to online LP and SDP covering. For online LP covering, they show a $O(1/(1-\lambda))$-consistent and $O(\log \frac{kn}{\lambda}$-robust algorithm, where $\lambda$ is a given parameter, $n$ is the number of variables, and $k$ is the upper bound for the ratio between the maximum positive entry and the minimum positive entry in each column of the constraint matrix. For online SDP covering, the same consistency and robustness are also obtained. If the variables in the (linear or semidefinite) program lie in a bounded region, the robustness can be improved slightly to $O(\log(s/\lambda)$, where $s$ is the sparsity.

**Questions:**

- The definition of ``consistency" in the paper is different from the standard one where the consistency refers to the competitive ratio when the prediction is accurate. In the context, the two definitions are the same because the optimal solution is predicted in the paper, and thus, ADV=OPT when the the prediction is accurate. However, in many other problems, even with an accurate prediction, ADV could be larger than OPT.

- The statements of algorithm 1 and algorithm 2 miss the update of the dual variables $y$. So when I first read the proof sketch, I got confused. The algorithms provided in the supplementary material are correct I guess.

**Limitations:**

The authors adequately addressed the limitations and I didn't see any potential negative societal impact.

**Strengths And Weaknesses:**

Strength:
- Online LP and SDP covering are very important frameworks in the field of online algorithmic design. They capture a large number of problems, such as ski rental, online set cover, and so on. The authors show a general framework to leverage the predicted (fractional) optimal solutions to design non-trivial algorithms. The work is solid and the proposed idea could be widely applied to many online covering problems.

Weakness:
- The robustness ratio matches the pure online worst-case bound only when the condition number $k$ is $poly(n)$, but in the general case, $k$ might be a very large value. The work can be strengthened if the paper has a discussion about the optimality of the consistency-robustness tradeoff. Ideally, prove that the dependence on $k$ is unavoidable when the consistency is $O(1/(1-\lambda))$. Furthermore, the authors can consider the case that $k$ is known in advance and see if a better ratio can be obtained.

---

> ### Author Response · Authors · 2022-08-02
> **Response to Reviewer LaFt**
>
> > The robustness ratio matches the pure online worst-case bound only when the condition number $\kappa$ is $\text{poly}(n)$, but in the general case, $\kappa$ might be a very large value. The work can be strengthened if the paper has a discussion about the optimality of the consistency-robustness trade-off. Ideally, prove that the dependence on $\kappa$ is unavoidable when the consistency is $O(1/(1-\lambda))$. Furthermore, the authors can consider the case that $\kappa$ is known in advance and see if a better ratio can be obtained.
>
> Thank you for the feedback. We note that we define $\kappa$ as the upper bound for the ratio between the maximum positive entry to the minimum positive entry in the matrix $A$ (see line $107$ right before Theorem $1.3$). If the matrix entries are polynomially bounded, which is a reasonable assumption, then $\kappa = \text{poly}(n)$. Otherwise, some non-zero entry of $A$ is exponentially small or large and cannot be represented efficiently, i.e., using $O(\log n)$ bits which is the standard assumption in the word RAM model of computation.
>
> Proving the consistency-robustness trade-off lower bound is one of the most intriguing future research problems. As stated in the conclusion section of the supplementary material, a potential direction is to follow the work by Wei and Zhang (NeuRIPS 2020), which showed the tight trader-off lower bounds between the robustness and consistency ratios for the ski rental and the non-clairvoyant scheduling problem. These problems are fairly structured, and the challenge is that the online covering LP/SDP problem we study is quite general. The current algorithms ensure consistency in a way that balances the contribution from the primal variables, which is where the condition number shows up in the robustness ratio. We believe this line of research is a promising road map.
>
> > The definition of ``consistency" in the paper is different from the standard one where the consistency refers to the competitive ratio when the prediction is accurate. In the context, the two definitions are the same because the optimal solution is predicted in the paper, and thus, ADV=OPT when the prediction is accurate.
>
> We note that we define an algorithm to be $C(\lambda)$ consistent if the cost of the online algorithm is bounded by $C(\lambda)$ times the cost of following the predictions blindly, even if the predictions are inaccurate. In particular, we $\text{\emph{do not}}$ assume that the advice is optimal (either online or offline optimal). While this could certainly be the case if the predictions are excellent, the predictions can also potentially be highly erroneous, especially if they are curated using an ML model. Thus we also provide robustness guarantees, which state that our algorithm's performance is also no worse than a $R(\lambda)$ factor of the offline optimal algorithm. The $R(\lambda)$ factor can be thought of as being approximately equal to the competitive ratio of the optimal $\text{\emph{online}}$ algorithm.
>
> Therefore we obtain the "best" of both worlds: if the predictions are excellent, we can obtain a better competitive ratio beyond what standard online algorithms without predictions are able to provide. In the case the hints are unhelpful, we have safeguards which ensure our algorithm's performance is still comparable to online algorithms without predictions.
>
> If we have not fully addressed this question, we would be more than happy to further elaborate on our consistency definitions.
>
>
> > The statements of algorithm 1 and algorithm 2 miss the update of the dual variables. So when I first read the proof sketch, I got confused. The algorithms provided in the supplementary material are correct I guess.
>
> Instead of bogging down the algorithmic description with the formal details of the dual variable updates, we chose to describe the primal variable update at an intuitive level in terms of its growth rate and defer the formal details of the dual variable updates to the proof sketch and the supplementary section. Specifically, the primal variables are exponential functions of the dual variables and we continuously increment the dual variables. As shown in Algorithms 2 and 6 in the supplementary material, these exponential functions are based on the "boundary coefficients", the cost vector in the objective, and the constraint coefficients. In particular, the boundary coefficients are based on the value of the dual variables and the advice. In the event that our paper is accepted, we will use the additional content page allowed for the camera-ready version to present the explicit primal-dual algorithms (Algorithms 2 and 6 in the supplementary material) in the main body.

---

> ### Author Response · Authors · 2022-08-07
> **Follow-up to Reviewer LaFt**
>
> Hi Reviewer LaFt,
>
> We were wondering if you had a chance to consider our responses to your initial review for our paper. We believe we have been able to satisfactorily address not only your questions but also the current concerns of all reviewers. If you agree that the questions you listed have all been addressed, could you please consider raising your score appropriately? If not, could you please let us know which concerns were not sufficiently addressed so that we have a chance to respond? Thanks!

---

> > ### Author Response · Authors · 2022-08-08
> > **Second Follow-up to Reviewer LaFt**
> >
> > Since the discussion period is ending soon, we just wanted to check again whether the rebuttal clarified and answered the initial questions raised in your review. We would be very happy to engage further if there are additional questions!
> >
> > Also, we wanted to check whether we could provide any additional clarification regarding the merits of the paper that would convince the reviewer to raise the score.

---

### Official Review · Reviewer_8aZd · 2022-07-11

**Rating:** 7
**Confidence:** 2
**Soundness:** 4 excellent
**Presentation:** 3 good
**Contribution:** 4 excellent

**Summary:**

An online optimization algorithm is a method where the data of the problem can be presented in a piecemeal fashion, and the optimizer must make certain irrevocable decisions at each step. This is a strong limitation, and usually results in a loss in performance described by the "competitive ratio" (ratio of online optimum to offline optimum).

The present paper belongs to a growing line of research where one studies online algorithms that receive "advice" on the solution. This advice may come from a predictive algorithm that guesses the next problem instances. The desiderata for a good method in this setting are the following: (1) if the advice is good, then the performance of the method should nearly match the optimal offline solution; and (2) if the advice is not good, the usual, assumption-free competitive ratio is (nearly) achieved.

In comparison with previous work, such as the one by Bamas et al. in NeurIPS 2020, the main novelties here are twofold. First, the present method can deal with certain "covering LPs" with non-integral constraints. Second, covering SDPs are also considered, which extends the scope of the "primal-dual method". In all cases, one obtains desiderata (1) and (2) above. A small experimental section is also presented.

Let me add a few words about the main ideas. As mentioned, Bamas et al. consider integral constraints. The present paper allows for fractional constraints; so decision variables must be increased continuously from round to round. The analysis of the method is not terribly difficult, but is a bit subtle and (at least to me) surprising.

**Questions:**

Can you comment a bit more on the motivation for the setting? E.g. where and when would it be a good idea to use these methods.

**Limitations:**

Yes, they have.

**Strengths And Weaknesses:**

*Strengths*

The paper is easy to read, clear, and seemingly original. The main issues at stake stated and the contribution is quite nice. Some previous results are subsumed by the present manuscript. While I am not an expert on this topic, I am fairly convinced that the extension of the primal-dual method to this setting is non-trivial.

*Weaknesses*

In practical applications, better control of the constants in the approximation ratios is needed.

---

> ### Author Response · Authors · 2022-08-02
> **Response to Reviewer 8aZd**
>
> > In practical applications, better control of the constants in the approximation ratios is needed.
>
> Empirically we observed that the competitive ratios were quite small, for example in our real graph datasets, they were bounded by $3$.
>
>
> > Can you comment a bit more on the motivation for the setting? E.g. where and when would it be a good idea to use these methods.
>
> In practice, good predictors can be learned for datasets with auxiliary information. Indeed, this is the premise of the emerging field of "learning-augmented / data-driven" algorithms. One compelling use case is if we are repeatedly solving the same algorithmic problem on related datasets, such as datasets which are time-varying such as the graph dataset used in our experiments. In such cases, we can simply use the solution given by an offline LP or SDP solver on a prior instance of the dataset as predictions. Therefore, predictors are readily and easily available for a wide class of natural datasets. Furthermore, classical online algorithms are overly pessimistic and it is conceivable that in many practical applications, future inputs to online algorithms can be predicted using ML methods, which can also serve as predictions. This new paradigm for online algorithms has been adopted for a wide array of fundamental online algorithms, including scheduling and caching. Theoretically, we give sample complexity bounds in Appendix A of the supplementary material which shows that predictions can be learned efficiently in the standard PAC learning setting. This gives an end-to-end recipe for designing learning-based algorithms for covering linear programs with fractional advice and covering SDP problems.

---

> ### Author Response · Authors · 2022-08-07
> **Follow-up to Reviewer 8aZd**
>
> Dear Reviewer 8aZd,
>
> We were wondering whether you had a chance to reviewer our responses to your initial assessment of our paper. We hope that we have been able to resolve not only any concerns you initially mentioned but also the current concerns of all reviewers. If you think there are any remaining concerns that have not been sufficiently addressed, could you please let us know so that we have a chance to respond? Thanks!

---

> > ### Comment · Reviewer_8aZd · 2022-08-08
> > **No questions/concerns right now**
> >
> > Thanks for your answer!

---

### Official Review · Reviewer_b8M5 · 2022-07-13

**Rating:** 6
**Confidence:** 2
**Soundness:** 3 good
**Presentation:** 2 fair
**Contribution:** 3 good

**Summary:**

This paper studies online covering linear and semidefinite programs, and proposed algorithms that incorporate models with prediction. The main technical results show that the proposal primal and dual learning-augmented algorithm can achieve both consistency (when the prediction is accurate) and robustness (when the prediction is inaccurate). The techniques work for both online linear programs and semidefinite programs. Some empirical evaluations are presented to demonstrate the performance of the proposed algorithmic framework.

**Questions:**

It is not good to say semidefinite programming is a unifying framework that generalizes quadratically-constrained quadratic programs (QCQPs). The authors may mean that convex QCQPs can be reformulated into a special SDP. However, a general QCQP can be highly non-convex and contain NP-hard instances as special cases.

**Limitations:**

It is fine to put some technical proofs to the extended version. But it is better to make the main text be more independent, not like an extended abstract.

**Strengths And Weaknesses:**

Strengths:
+ The problem is well-motivated, and the contributions seem novel compared to the previous results.
+ The technical proofs in the extended version appear to be solid (but the reviewer didn't check the details).
+ The comparison between online linear programs and online semidefinite programs is well-organized.

Weaknesses:
- The main results have not been rigorously stated in the main text. Indeed, the informal results in Theorem 1.3 and Theorem 1.4 are not rigorous. The authors put the full algorithms, formal statements of theorems, and all the proofs into the extended version. The main text appears less complete and is more like an extended abstract.
- The overview of techniques in Section 1.2 is not easy to follow. The overview still appears to be very technical and has many jargons.
- The proof sketch for Theorem 1.3 and Theorem 1.4 on Page 7 and Page 8 does not present sufficient proof ideas, which is not easy to follow.

---

> ### Author Response · Authors · 2022-08-02
> **Response to Reviewer b8M5**
>
> > The main results have not been rigorously stated in the main text. Indeed, the informal results in Theorem 1.3 and Theorem 1.4 are not rigorous. The authors put the full algorithms, formal statements of theorems, and all the proofs into the extended version. The main text appears less complete and is more like an extended abstract. The proof sketch for Theorem 1.3 and Theorem 1.4 on Page 7 and Page 8 does not present sufficient proof ideas, which is not easy to follow.
>
> At a high level, our proof works by showing both robustness and consistency. For robustness, we increment the variables in terms of their growth rates by using the dual variables as a proxy. We then use the weak duality theorem and argue approximate dual feasibility to show robustness. For consistency, we argue that the change of the objective credited to the online algorithm is within an $O(1/(1-\lambda))$ factor of the objective credited to the advice (see the proof of Theorems 2.1 and 3.1 in the supplementary material for the details). Are there specific parts of the proof sketches that we can help clarify for your understanding? In the event that the paper is accepted, we would be permitted an additional content page, which we will use to provide more explanation of the intuition behind the proof sketches to make them easier to follow.
>
> > The overview of techniques in Section 1.2 is not easy to follow. The overview still appears to be very technical and has many jargons.
>
> Thanks for the feedback. Indeed, we provide a gentle but thorough introduction to specific terminology in Sections 1.1 and 1.2 of the full version in the supplementary material. We will make an effort to move these discussions to the main body to facilitate a better understanding of specific jargon, space permitting. We would be more than happy to provide any further clarifications if there are any specific technical points or terminology that the reviewer feels are not clarified in the text.
>
> > It is not good to say semidefinite programming is a unifying framework that generalizes quadratically-constrained quadratic programs (QCQPs). The authors may mean that convex QCQPs can be reformulated into a special SDP. However, a general QCQP can be highly non-convex and contain NP-hard instances as special cases.
>
> Thank you for pointing this out. Indeed, we meant that some convex QCQPs can be reformulated into our SDP formulation. We have clarified this in the revised version.

---

> ### Author Response · Authors · 2022-08-07
> **Follow-up to Reviewer b8M5**
>
> Hi Reviewer b8M5,
>
> We were wondering whether you had a chance to reviewer our responses to your initial assessment of our paper. We do think we are able to satisfactorily address not only your concerns but also the current concerns of all reviewers. If you agree that the points you mentioned have all been addressed, could you please consider raising your score appropriately? If not, could you please let us know which concerns were not sufficiently addressed so that we have a chance to respond? Thanks!

---

### Author Response · Authors · 2022-08-02
**Thanks to all reviewers**

We thank the reviewers for their thoughtful comments and valuable feedback. We especially appreciate the positive remarks, such as

* The problem is well-motivated, and the contributions seem novel compared to the previous results (Reviewer b8M5)

* The technical proofs in the extended version appear to be solid (Reviewer b8M5)

* The comparison between online linear programs and online semidefinite programs is well-organized (Reviewer b8M5)

* The analysis of the method is...subtle and (at least to me) surprising (Reviewer 8aZd)

* The paper is easy to read, clear, and seemingly original (Reviewer 8aZd)

* The main issues at stake stated and the contribution is quite nice (Reviewer 8aZd)

* The work is solid and the proposed idea could be widely applied to many online covering problems (Reviewer LaFt)

We provide our responses to the specific questions of each reviewer below. We hope our answers resolve all initial questions and concerns raised by the reviewers and we will be most happy to answer any remaining questions!

---

### Meta-Review · Area_Chair_nUjM · 2022-08-23

**Recommendation:** Accept
**Confidence:** Certain

**Metareview:**

Thank you for your submission to NeurIPS. The reviewers unanimously found the work to address an important, relevant problem, and the paper to be clear and well-written. All three reviewers unanimously recommend accepting the paper.

Please incorporate reviewer feedback in preparing the camera ready version. In particular, please take care to incorporate the clarifications outlined in your response to Reviewer b8M5.

**Award:**

No

---

### Decision · Program_Chairs · 2022-09-14

Accept